# ProTargetMiner as a proteome signature library of anticancer molecules for functional discovery

Amir Ata Saei [1], Christian Michel Beusch [1,7], Alexey Chernobrovkin [1,2,7], Pierre Sabatier [1,7], Bo Zhang [1,3,7], Ülkü Güler Tokat[1,4], Eleni Stergiou [1], Massimiliano Gaetani [1,5], Ákos Végvári [1] & Roman A. Zubarev[1,5,6]*

Deconvolution of targets and action mechanisms of anticancer compounds is fundamental in drug development. Here, we report on ProTargetMiner as a publicly available expandable proteome signature library of anticancer molecules in cancer cell lines. Based on 287 A549 adenocarcinoma proteomes affected by 56 compounds, the main dataset contains 7,328 proteins and 1,307,859 refined protein-drug pairs. These proteomic signatures cluster by compound targets and action mechanisms. The targets and mechanistic proteins are deconvoluted by partial least square modeling, provided through the website http://protargetminer.genexplain.com. For 9 molecules representing the most diverse mechanisms and the common cancer cell lines MCF-7, RKO and A549, deep proteome datasets are obtained. Combining data from the three cell lines highlights common drug targets and cell-specific differences. The database can be easily extended and merged with new compound signatures. ProTargetMiner serves as a chemical proteomics resource for the cancer research community, and can become a valuable tool in drug discovery.

[1] Department of Medical Biochemistry and Biophysics, Karolinska Institutet, 171 77 Stockholm, Sweden. [2] Pelago Bioscience AB, 171 48 Solna, Sweden. [3] Department of Biosciences and Nutrition, Karolinska Institutet, 141 86 Stockholm, Sweden. [4] Faculty of Science, Department of Chemistry, Hacettepe University, 06 800 Ankara, Turkey. [5] SciLifeLab, SE-17 177, Stockholm, Sweden. [6] Department of Pharmacological & Technological Chemistry, Sechenov First Moscow State Medical University, Moscow, Russia 119146. [7] These authors contributed equally: Christian Michel Beusch, Alexey Chernobrovkin, Pierre Sabatier, Bo Zhang. *email: Roman.Zubarev@ki.se

Deciphering the targets, mechanism of action (MOA) and cellular effects for compounds, especially for those derived from phenotypic screenings, are all indispensable and challenging tasks in drug discovery[1]. These tasks can be addressed by connecting the affected cellular phenotypes to small molecules by connectivity maps[2–7]. Such approaches explore the similarity of the cell response signature produced by a compound of interest with signatures of other compounds in the database. However, majority of connectivity map studies are based on gene expression profiles. Since proteins are the targets of most drugs, proteome responses can be more specific to drug action. Only one recent effort has reported a connectivity map based on phosphoproteomic and chromatin signatures, measuring 100 phosphopeptides and 59 histone modifications for treatments with 90 drugs[8], but such focused signatures might not be relevant for all compounds. In principle, protein abundances should serve a good basis for connectivity maps; after all, biological systems are defined by their proteome state. Also, abundances of proteins are as much determined by their degradation as by expression[9,10], both reflected uniquely in proteomics data. Indeed, in several studies, no strong correlation between mRNA levels and protein concentrations exists even at the steady state[11]. In dynamic situations where degradation processes play an important role, such as programmed cell death[12], the relationship between the transcriptome and proteome may become even less direct.

Here we use chemical proteomics to study the relationship between the anticancer drug molecules and the dying cell phenotypes induced by these molecules[13]. Chemical proteomics has traditionally been defined as the use of small molecules (which are considered known entities) in studying the unknown functions of proteins[14]. Recently, chemical proteomics began to designate also the opposite approach, in which proteome analysis is applied to studying functions of small molecules[13,15–18].

We have previously shown that when sensitive cell lines are treated with a toxic compound, its targets and mechanistic proteins are consistently found among the most regulated ones; mapping these proteins on known networks can reflect the compound MOA. This observation served as a basis for the chemical proteomics method called Functional Identification of Target by Expression Proteomics (FITExP)[19]. In many cases, the affected target and other mechanism-related proteins are found up-regulated. This can be explained by a feedback effect when inhibition of certain proteins activates their (over)production[20]. The alternative effect, involving target protein down-regulation, can be caused by, e.g., protein proteolysis[21]. Such feedback phenomena have also been documented in more primitive organisms[22,23].

The classic FITExP experiment increases the specificity for a given compound by adding a panel of other molecules. The specificity parameter in FITExP reflects the protein regulation for a given compound compared to regulation by other molecules. Therefore, proteins specifically responding to a compound of interest can be identified. Using the specificity parameter, FITExP could successfully identify the targets of several chemotherapeutics[19], probe the targets and MOA of metallodrugs[24] and even toxic nanoparticles[25]. We have also shown that combining the proteomics data from treated matrix-attached and matrix-detached cells can improve the deconvolution of drug targets and MOAs[17]. Achieving a high level of specificity in analysis usually requires the use of several compounds and cell lines. We hypothesized that the equivalent increase in specificity can be obtained in a single-cell line with a multitude of contrasting compounds.

Here, we profile 56 compounds at LC50 concentrations and show that contrasting the proteome signature of a single compound against others highlights a given compound's target and mechanistic proteins on the top positions in most cases.

Furthermore, we show that the contrasting panel for a single-cell line can be reduced to 8 compounds. With this miniaturization opportunity, using 9 molecules representing orthogonal MOAs, deep proteome datasets are obtained for 3 major cancer cell lines: MCF-7, RKO, and A549. When the data from the three cell lines are combined, common targets and MOAs are revealed, while investigation of the differences highlights important cell-specific mechanisms. The ProTargetMiner database is expandable using the provided user interface that, in turn, is modifiable as it is written in R Shiny. The input is the fold changes of proteins in a number of replicates for a given compound in the desired cell line(s), and the output is an interactive PLS-DA model with extractable rankings, providing the likely drug target and MOA. ProTargetMiner is also directly available through http://protargetminer.genexplain.com.

## Results

**Overview.** Here we present the ProTargetMiner concept. The overview of the project's objectives is given in Fig. 1a. Employing the specificity concept (Fig. 1b), orthogonal partial least square-discriminant analysis (OPLS-DA) modeling (Fig. 1c) contrasts the proteome signature of a given compound against those of the rest of the compounds, which reveals the compound targets, MOA, effects on protein complexes and potential resistance factors. The workflow is given in Fig. 1d.

We selected A549 human lung adenocarcinoma cell line as a model system, because it is well covered in literature, and showed the highest sensitivity to selected compounds among the tested cell lines (MCF-7 and RKO). Viability measurements were performed for 118 clinical anticancer molecules selected from Selleckchem FDA-approved drug library together with several experimental anticancer compounds with unknown targets. A collection of 56 compounds with LC50 below 50 μM was chosen to treat the cells (at LC50 concentrations) for 48 h in three replicates. With the biological effect (cell death) being of the same magnitude, the differences in the proteome states could be attributable to the differences in targets and MOAs. This is in contrast to other databases, where fixed concentrations were used. The selected compounds belong to 19 different classes with versatile targets and MOAs, spanning 112 known targets curated from DrugBank (https://www.drugbank.ca/) in November 2019. These compounds and their known targets are listed in Supplementary Table 1. As standard drugs used for quality control, methotrexate, paclitaxel, and camptothecin were chosen and included in each TMT10 multiplexed proteomics experiment (labeling information is given in Supplementary Table 2).

For the main dataset, 287 proteomes were analyzed (10 conditions (compounds+controls) in 5 replicates in the first experiment+79 conditions in 3 replicates). Overall, to obtain the main dataset, 229 LC-MS/MS analyses were performed after multiplexing and fractionation. In total, 144,075 peptides were quantified, being attributed to 7,328 proteins with at least 2 unique peptides per protein. After selecting only proteins quantified with no missing values for 50 drugs, the list was reduced to 4,557 proteins (Supplementary Data 1) that were used in all subsequent analyses.

In each of the 9 multiplexed experiment for the original dataset, the compounds methotrexate, camptothecin and paclitaxel were included as controls, so that they can be used for data quality check. The Pearson correlation coefficient $r$ for the average normalized intensities for the above drugs in different experiments was between 0.859 and 0.995 (only proteins with no missing values were used in this analysis), attesting to the quality of the proteomics data (Supplementary Fig. 1).

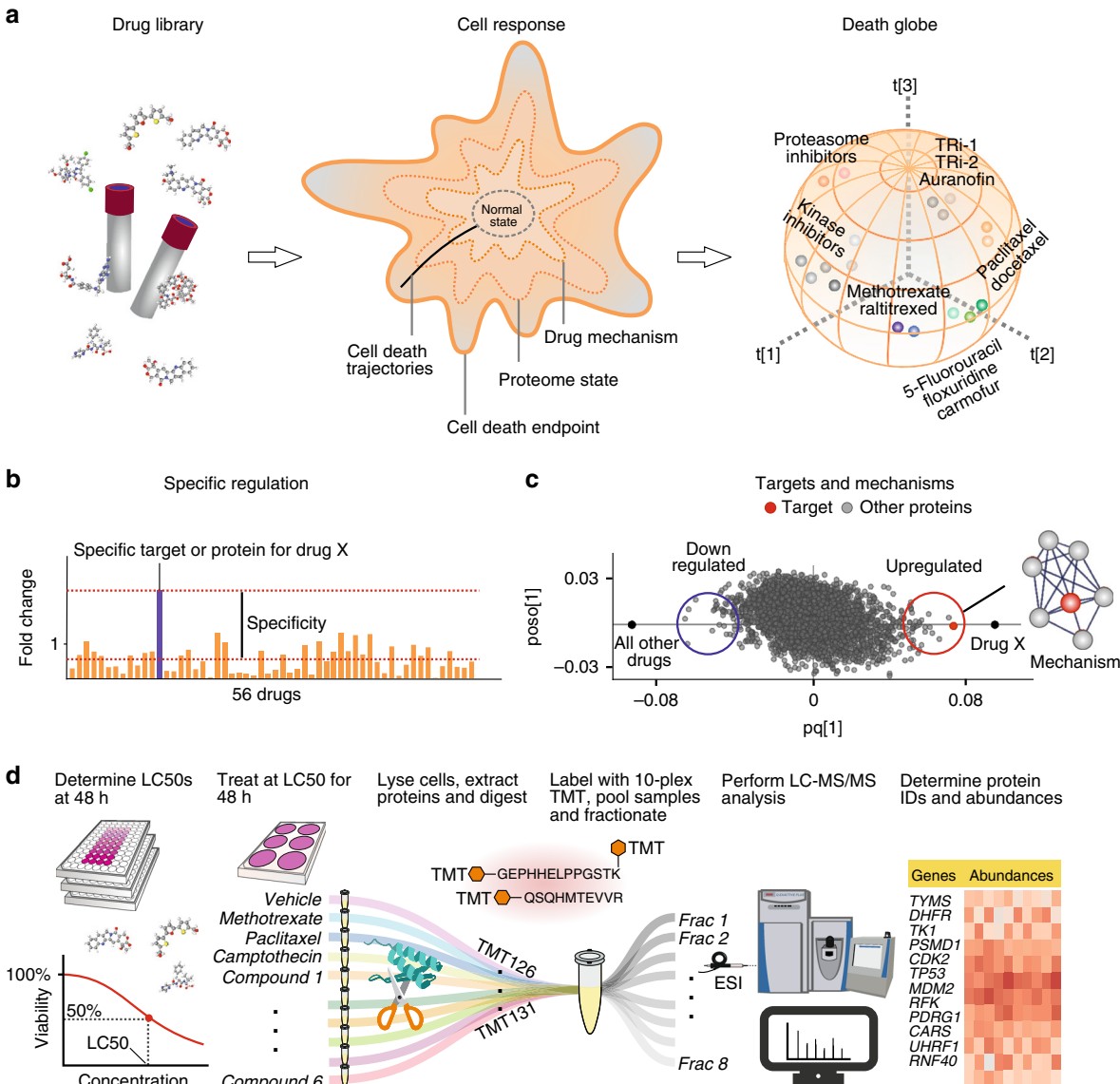

**Fig. 1 ProTargetMiner strategy and workflow. a** an extensive proteome signature database of anticancer molecules will identify compounds with similar MOA in multidimensional space. **b** since the response of the target and mechanistic proteins to a compound is specific, **c** an OPLS-DA model contrasting the given compound with all other molecules in the database will identify the drug targets (red circles) and mechanistic proteins as specifically regulated proteins (among other proteins shown with gray circles). **d** workflow: determination of LC50 values for the library of compounds and selected cells; cell treatment with 56 compounds as well as vehicle-treated control and standard treatments (methotrexate, paclitaxel and camptothecin) in at least three biological replicates; lysis, digestion and labeling with TMT-10plex reagents; multiplexing the 10-plexed samples; fractionation of the pooled sample to increase the proteome coverage; analysis of individual fractions by LC-MS/MS; protein identification and relative quantification; data post-processing. The compound structures in panel a were taken from PubChem.

Due to the nature of random sampling of peptides in shotgun proteomics, the missing values cumulatively increase by merging several datasets, as not all proteins are quantified in all 9 experiments. The comparison of number of proteins, number of peptides, average sequence coverage and the number of missing values for the 9 experiments as well as for the merged original dataset is given in Supplementary Fig. 2.

**Compound clusters, protein clusters, and their interactions**. To reduce the number of dimensions and visualize the proteomic space, we employed a nonlinear dimension reduction method t-SNE that is widely used for projection of multidimensional molecular signatures[26]. On the resultant 2D Death map, where the drug-induced proteome signatures are mapped as points

(Supplementary Fig. 3), we used the proximity of these points to evaluate the similarity of the drug-induced signatures. As expected, drugs with similar MOAs (e.g., tubulin inhibitors paclitaxel, docetaxel, vincristine, and 2-methodyestradiol; pro-teasome inhibitors b-AP15 and bortezomib[27]; pyrimidine analogs 5-fluorouracil, floxuridine and carmofur; thioredoxin reductase 1 (TXNRD1) inhibitors auranofin, TRi-1 and TRi-2[28]; and DNA topoisomerase 1 (TOP1) inhibitors camptothecin, topotecan and irinotecan) were proximate on the t-SNE plot, confirming that the Death map can be used for evaluating the MOA similarities.

We found tomatine to be a gross outlier in principal component analysis (PCA) (Supplementary Fig. 4a). For tomatine, the total number of differentially regulated proteins with 1.5 and 2 fold cutoffs (vs. control) compared to the average of all other drugs was 9.4 and 14.6 fold higher, respectively. In

Supplementary Fig. 4b, the number of differentially regulated proteins (fold change vs. control >2 and <0.5) for tomatine vs. other compounds is shown. Tomatine is likely to act via proteasome inhibition[29], along with unspecific membrane damage[30]; these effects may explain the extraordinary changes induced by tomatine in the cell proteome. Therefore, we excluded tomatine from subsequent analyses.

PCA revealed 14 orthogonal dimensions contributing at least 1% to separation of proteome signatures (excluding tomatine) (Supplementary Fig. 5). The first 3 components are shown in Supplementary Fig. 6.

We next employed a conventional correlation-based hierarchical clustering analysis, in which the compounds aggregated in clusters mostly based on common targets/MOA (Fig. 2a). There are two super-clusters separating the compounds: one composed of the compounds that directly or indirectly lead to DNA damage, such as pyrimidine analogs, as well as TOP1 and TOP2 inhibitors, and the second super-cluster containing all the other molecules. The second super-cluster is in turn divided into proteasome inhibitors and the rest of molecules. This can be explained by dramatic accumulation of misfolded proteins or proteotoxicity of proteasome inhibitors[31,32], which is not the case with any other compound class. For example, for bortezomib the number of up-regulated proteins was much higher than down-regulated proteins (up/down ratio of 17.8 for bortezomib (vs. control) compared to the average of 2.9 for all other drugs at a minimum regulation of 1.5 fold). The ranking of drugs by the overall deviation of their molecular signatures from the untreated state is shown in Supplementary Fig. 7.

It must be noted that the a priori annotation of compounds is solely based on anticipated targets and disregards the off-targets effects, while proteome-based clustering is based on the overall change of the proteome. For example, auranofin clusters with b-AP15, consistent with its recently identified deubiquitinase inhibitor activity[33,34]. Note that kinase inhibitors, although seemingly diverse in their cellular effects, also showed a fair degree of clustering.

We also performed clustering of proteins and identified 15 clusters (vertical axis in Fig. 2a), subjecting each cluster to Gene Ontology analysis (Supplementary Data 2 and Fig. 2b). Some of these clusters represent high density protein networks: e.g., cluster 13 maps to ribosome.

A quick look at the heat map in Fig. 2a reveals protein clusters due to which the compounds are placed in specific clusters. For example, the compounds in super-cluster 1 are separated from super-cluster 2 mostly due to the differences in protein clusters 6 and 15, which represent chromosome condensation and p53 signaling pathways, respectively. The DNA damaging agents lead to induction of p53 signaling pathway resulting in cell cycle arrest[35]. As another example, pyrimidine analogs 5-fluorouracil, floxuridine and carmofur form a mini-cluster because they down-regulate ribosomal proteins in the protein cluster 13[36].

Radar charts on Fig. 2c–i visualize the engagement of compound groups in different protein clusters. For example, proteasome inhibitors (and also everolimus) strongly induce cluster 12 related to protein folding (Fig. 2c), and so does auranofin, unlike other TXNRD1 inhibitors (Fig. 2d). Unexpectedly, some kinase inhibitors including lapatinib, bosutinib, sunitinib, and gefitinib up-regulate the (chole)sterol synthesis pathways, represented by cluster 14 in Fig. 2e. We later verify that these kinase inhibitors indeed enhance cellular cholesterol levels. TOP1 and TOP2A inhibitors down-regulate cluster 6 related to chromatin condensation (Fig. 2f, g). Also down-regulated are the ribosomal proteins (cluster 13) by pyrimidine analogs as well as oxaliplatin[37] (Fig. 2h) and mitochondrial pathways (cluster 2) by a group of mitochondria-targeting compounds (Fig. 2i).

**Functional discovery at the protein level**. Protein regulation is usually defined as a ratio of the protein abundances in the cells incubated with a drug and a vehicle (control). However, many proteins are involved in generic, drug-unspecific cell responses (e.g., detoxification, death, or survival pathways). To reveal the protein responses characteristic to a particular drug, FITExP introduced specificity as a ratio of the protein regulation in response to a particular drug to the median regulation in response to all other drugs, and used it for deconvolution of drug target/MOA[19]. Here, we merge the regulation and specificity into one parameter by employing OPLS-DA[38], contrasting a given compound against all other molecules (Fig. 3a).

OPLS-DA is a multivariate data analysis tool and a supervised modeling approach that is useful for highlighting what makes two groups or systems different[38]. In ProTargetMiner, OPLS-DA is used for discriminant analysis. OPLS-DA would thus discover variables (here proteins) with the largest discriminatory power. The two-group comparison models are easiest to interpret, because there will be only one predictive component. This predictive component is rendered in the x axis in the loading or score scatter plot. Therefore, the horizontal axis in the score scatter plot will demonstrate the variation between the groups, while the vertical dimension and any higher orthogonal components will capture variation within the groups (the latter is quite unimportant in this study).

In thus obtained OPLS-DA models, where each protein is represented by a dot on a loading plot, the proteins specifically up- or down-regulated in response to a given treatment are found on the opposite sides of the plot on the x axis (Fig. 3b). The proximity of a protein to the endpoint on either side of the x-axis reflects the magnitude and specificity of regulation of that protein in response to a given drug, taking into account the variation among the replicates. Since the y-axis coordinate reflects the contribution of the orthogonal components, the most specific target candidates are located near the x-axis.

Each OPSL-DA model is characterized by R2 value representing the goodness of the model fit. To avoid overfitting of the data in multivariate analysis, SIMCA employs leave-one-out cross-validation strategy. The result is the Q2 value, which is a measure of model predictive power. Q2 is also called the cross-validated R2 and should obviously have a smaller value. Therefore, a model with R2 of 1 perfectly describes the data, and the Q2 value of 1 indicates perfect predictivity of the model. In brief, for cross validation, the whole data-set is divided into seven groups and seven subsequent models are developed based on 6/7 of the data, leaving a new group aside each time[39]. The deleted data are then used as a verification set, and the differences are calculated between the actual and predicted values. These differences are normalized and subtracted from unity, which provides the Q2 value. For more detailed explanation, see Umetrics documentation [39].

As representative examples of drug target deconvolution, OPLS-DA models for several drugs are shown in Fig. 4. R2 and Q2 values are given on each plot. The methotrexate target dihydrofolate reductase (DHFR) is convincingly identified as an obvious up-regulated outlier (p vs. control = 2E-5, two-sided Student's t-test) (Fig. 4a). In SIMCA software, variable influence on projection (VIP) values can also be extracted from the model loading[40], which show the total contribution of x variables to the OPLS-DA model. These VIP values are summed over all components and weighted with regards to the Y variation explained by each component. Therefore, VIP values can be used for ranking of target proteins. As an example, the VIP values indicate that DHFR is the second most contributing protein to methotrexate treatment (Fig. 4b). Network analysis of the specifically regulated proteins on the either side of the model

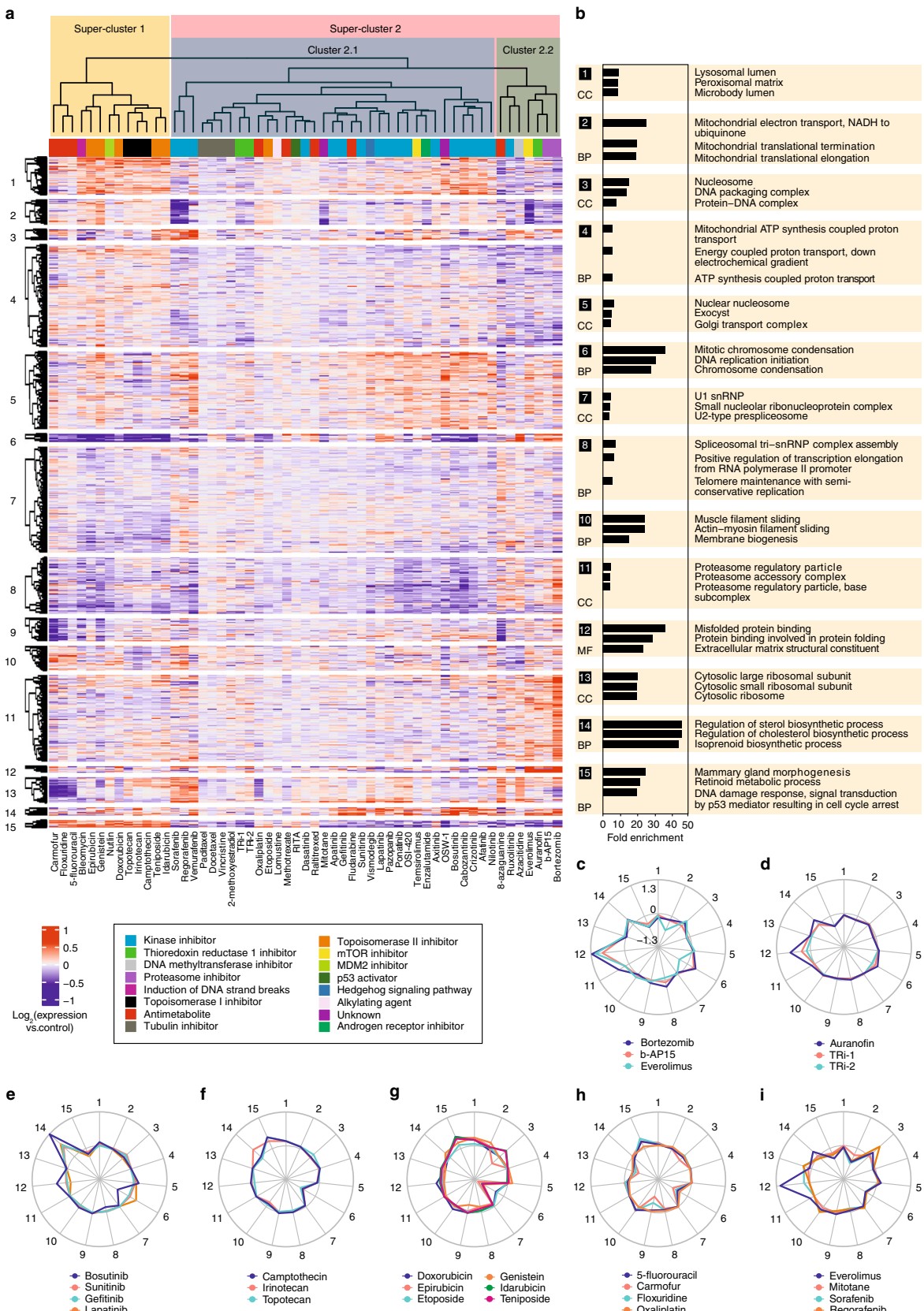

**Fig. 2 Hierarchical clustering of the proteome signatures by compounds and proteins. a** compound clustering is largely consistent with their presumed MOAs (compound classes are denoted by colors). **b** top three most enriched gene ontology terms, representing either molecular function (MF), biological process (BP), or cellular component (CC). Cluster 9 that did not map to any specific GO term is omitted. **c–i** radar plots showing the clusters targeted by representative compounds. The distance from the center is proportional to the mean logarithm of fold change, and inner circle represents zero regulation. Panel **a** was made from Supplementary Data 1. Source data are provided as a Source Data file.

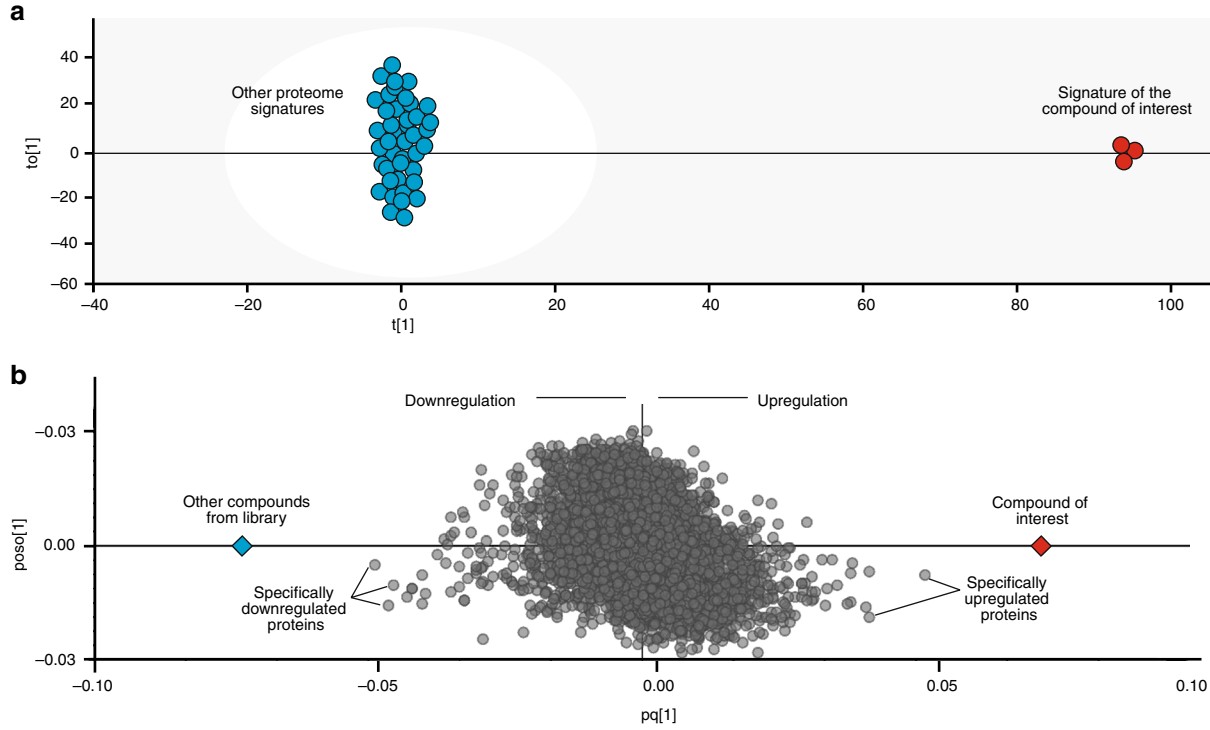

**Fig. 3 OPLS-DA paradigm. a** a generalized OPLS-DA model contrasting a given compound proteome signature against all others. **b** the OPLS-DA loading or score scatter plot demonstrating proteins most contributing to class separation.

reflects compounds' MOA. As an example, processes enriched for 30 top specifically up-regulated proteins in methotrexate treatment are deoxyribonucleoside monophosphate biosynthetic process ($p < 1E\text{-}6$) and tetrahydrofolate metabolic process ($p < 7E\text{-}6$) and the enriched function is folic acid binding ($p < 5E\text{-}5$), in line with the known drug mechanism.

Tubulins are found to be the most specifically up-regulated proteins for paclitaxel and down-regulated for vincristine, consistent with these two drugs affecting tubulin depolymerization[41] and polymerization[42], respectively (Fig. 4c, d).

If mechanistic proteins are engaged in large complexes, whole complexes can be specifically regulated. For example, the proteasome inhibitor bortezomib demonstrates specific up-regulation of the proteasome subunits (Fig. 4e). The sorafenib model shows specific down-regulation of NADH dehydrogenases and mitochondrial ribosomal proteins (Fig. 4f). This latter finding is in line with an earlier report for human neuroblastoma cells[43] and shows that ProTargetMiner results can be cautiously generalized to other cell lines. However, due to the lower depth of the original dataset and the low abundance of kinases, among the known sorafenib targets, only RAF1 data was available in the dataset and this protein was not among the top proteins in OPLS-DA model. We later discuss why ProTargetMiner may not be the tool of choice for target deconvolution of kinase inhibitors.

The OPLS-DA derived x coordinates (specificity values) of each protein to each of the 55 tested compound are provided in Supplementary Data 3, and can serve as a reference resource in other studies, along with the expression data already presented in Supplementary Data 1.

The expression levels of the top proteins for different compounds are shown in Fig. 4g–j. For example, Fig. 4g shows the higher expression of tubulins identified through OPLS-DA in paclitaxel treatment vs. the average expression for all other drugs.

To further demonstrate the validity of the approach and to show what happens when no true signal is present, we removed the compounds-related columns in Fig. 4 from the dataset one by

one (3 cases) and built OPLS-DA models with three randomly chosen (irrelevant) columns instead. The protein targets highlighted in Fig. 4 disappeared from the top ranking list, indicating that random selection of columns does not support meaningful findings (Supplementary Fig. 8).

**Functional discovery on kinase inhibitors.** As shown in Fig. 2, lapatinib, gefitinib, and other kinase inhibitors such as bosutinib, sunitinib, crizotinib, and cabozantinib affect cholesterol metabolism and/or lipid synthesis (representative OPLS-DA models and up-regulated proteins in Supplementary Fig. 9a, b, respectively). Literature seems to support these results. Lapatinib and crizotinib can induce cholesterol accumulation in human cardiomyocytes[44]. Lapatinib can also induce the accumulation of cholesterol in late endosomes in breast cancer cells[45]. Increased cholesterol levels is a common side effect of cabozantinib in clinical trials. Furthermore, new-onset hyperlipidemia has been noted in patients taking sunitinib[46] and in mice in vivo[47].

To experimentally verify the effect of kinase inhibitors on cellular cholesterol levels, lapatinib, bosutinib, sunitinib, and gefitinib were tested, while sorafenib that did not induce up-regulation of lipid synthesis and/or cholesterol metabolism proteins, was used as a negative control. To avoid excessive cytotoxicity, we used a shorter incubation time of 20 h and a sub-LC50 concentration (4 μM) for all the compounds. A549, RKO, and human foreskin fibroblast (HFF-1) cells were used for the analysis. While all compounds increased the cellular total cholesterol levels by ~20–50% in A549 cells, in RKO cells, only lapatinib, gefitinib, and bosutinib had a similar effect, and in HFF-1 cells, only lapatinib and bosutinib enhanced the cholesterol levels. In short, the prediction made by ProTargetMiner was confirmed (Supplementary Fig. 9c). The potential contribution of this effect on the cytotoxicity of kinase inhibitors can be a subject for future studies.

Biochemical pathways affected by compounds are related not only to death pathways but also to cell survival[17]. Therefore, the

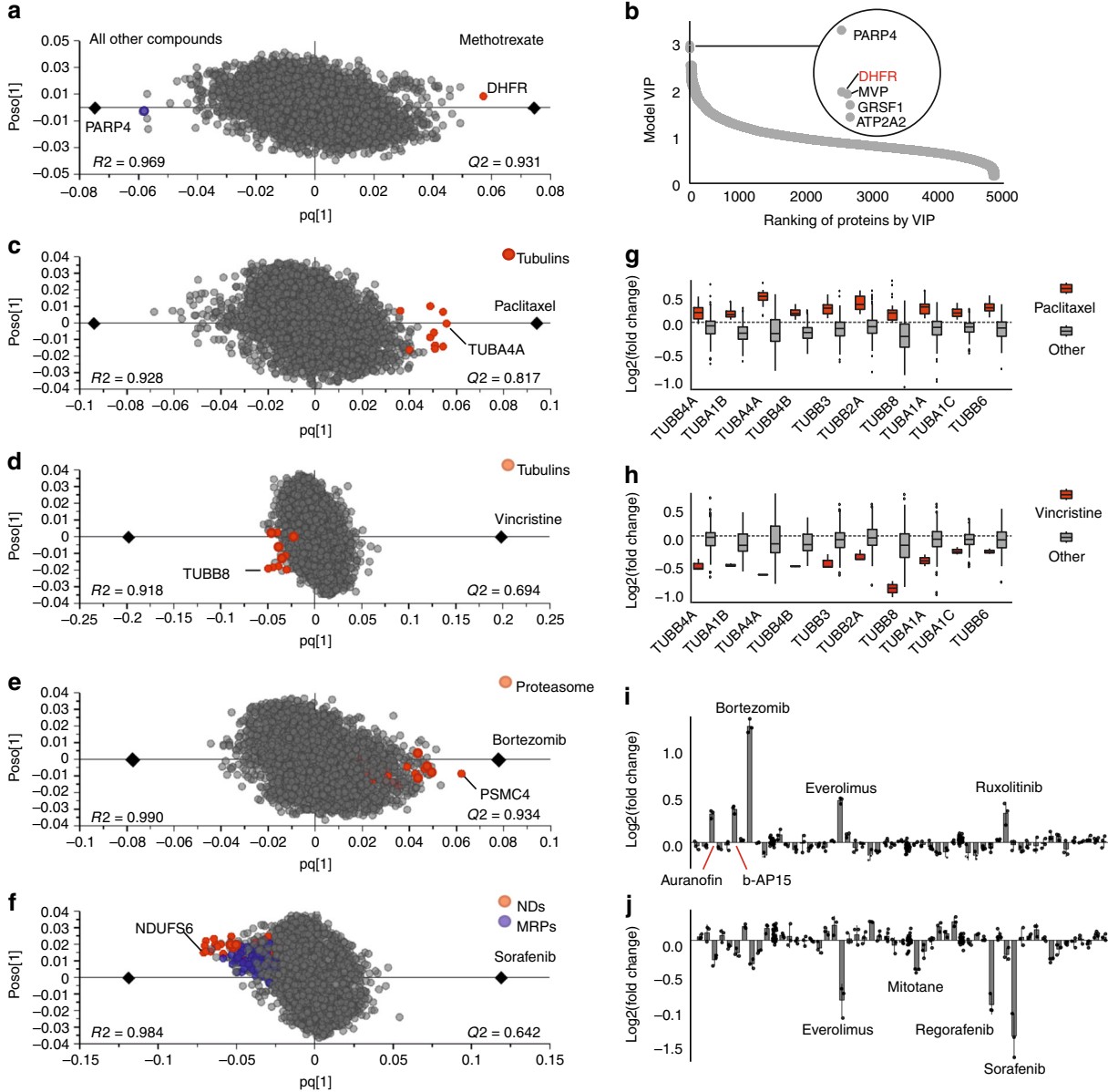

**Fig. 4 ProTargetMiner reveals drug targets, action mechanisms, and affected cellular complexes. a** OPLS-DA model for methotrexate. **b** variable influence on projection (VIP) values extracted from methotrexate OPLS-DA model. **c–f** OPLS-DA models for four other representative compounds (drug targets and/or mechanistic proteins are shown in red and dark blue circles). The mechanistically relevant pathway enrichment for the 30 most specifically up- or down-regulated proteins: GO processes for paclitaxel - mitotic cell cycle ($p < 9E-11$) and microtubule-based process ($p < 8E-8$); vincristine - mitotic cell cycle ($p < 1E-5$) and microtubule cytoskeleton organization ($p < 2E-4$). Microtubule was the top enriched component for both paclitaxel and vincristine ($p < 0.001$). Proteasome complex ($p < 4E-7$) and NADH dehydrogenase complex (in red, $p < 7E-21$) were the top components enriched for bortezomib and sorafenib, respectively. Mitochondrial translation processes (in blue, $p < 0.001$) were also enriched in GO terms for sorafenib (selection of 30 top proteins in either side of x axis is empirical and leads to less redundancy in pathways). MRP = mitochondrial ribosomal proteins, ND = NADH dehydrogenase. **g** the regulation of tubulins in response to paclitaxel vs. control in comparison with all other compounds. **h** the regulation of tubulins in response to vincristine vs. control in comparison with all other compounds (Center line, median; box limits contain 50%; upper and lower quartiles, 75 and 25%; maximum, greatest value excluding outliers; minimum, least value excluding outliers; outliers, more than 1.5 times of upper and lower quartiles). **i** the regulation of PSMC4 in response to bortezomib vs. control in comparison with all other compounds. **j** the regulation of NDUFS6 in response to sorafenib vs. control in comparison with all other compounds. Data are represented as mean ± s.d. ($n \geq 3$ biologically independent experiments). Panels **a** and **c–f** were made from Supplementary Data 1). Source data are provided as a Source Data file.

specifically regulated proteins could be potentially linked to drug resistance. For example, EGFR that was specifically up-regulated in the sorafenib model (Supplementary Fig. 10a) is known to be involved in resistance to this drug[48]. Another kinase specifically up-regulated in response to sorafenib (and regorafenib) was AXL (Supplementary Fig. 10a). AXL is a receptor tyrosine kinase

regulating many aspects of cell proliferation and survival, and its overexpression induces resistance to EGFR targeted therapies[49]. When we combined sorafenib and regorafenib (at LC50) with the specific AXL inhibitor TP0903 in non-cytotoxic concentrations (<100 nM), the combination treatment significantly increased cell death in A549 cells at 24 h and 48 h compared to pure sorafenib

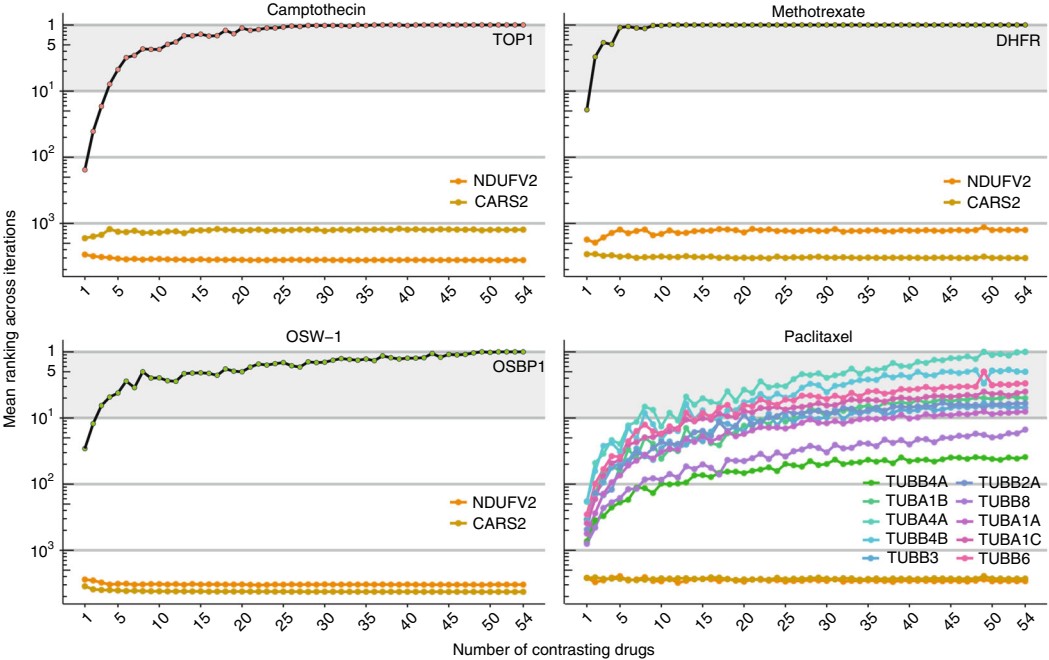

**Fig. 5 Determination of the ProTargetMiner minimal size. Four compounds were contrasted against 50 random combinations of 1–54 compounds by PLS-DA modeling and the mean drug target ranking was calculated for each number.** NDUFV2 and CARS2 proteins were randomly chosen as non-target proteins. Supplementary Data 1 was used for production of this figure.

and regorafenib (Supplementary Fig. 10b), validating the ProTargetMiner prediction, that AXL might induce resistance to these drugs.

**On the minimum size of ProTargetMiner.** Since compound-induced effects can be cell-specific, detailed characterization of drug action is desirable in the most relevant biological setting. Thus in the drug development process it would be advantageous to build a ProTargetMiner dataset with a customized drug panel and cell type. Creating a ProTargetMiner de novo, however, could be time-consuming and expensive. Miniaturization of the experiment requires determination of the minimal compound panel size for deducing the target and MOA. To address this issue, PLS-DA models were built for different numbers of contrasting compounds ($n = 1$–54, 50 molecule combinations randomized for each $n$). The mean rankings of the known targets for representative compounds camptothecin, methotrexate, OSW-1 and paclitaxel were determined for each number of contrasting drugs $n$. As expected, higher $n$ gave better ranking for drug targets but not for random proteins (Fig. 5). Encouragingly, already 8–10 contrasting molecules in the drug panel were in most cases enough for target rankings reaching a value below 10. This would present an opportunity to miniaturize ProTargetMiner for specialized applications, as well as reducing the labor and cost of the analyses.

**ProTargetMiner size vs. proteome depth.** A miniaturized Pro-TargetMiner dataset could offer a deeper proteome coverage with less missing values. We obtained deeper proteomic datasets for A549, MCF-7, and RKO cells representing major cancer types (lung, breast, and colon cancers, respectively) (Supplementary Data 4–6). As a drug panel, 9 molecules were chosen representing most diverse MOAs according to drug clustering in Fig. 2a as well as different orthogonal dimensions in the factor analysis of that dataset: 8-azaguanine (target: PNP), raltitrexed (target: TYMS), topotecan (target: TOP1), floxuridine (target: TYMS), nutlin (target: MDM2), dasatinib (target: multiple kinase targets), gefitinib

(target: EGFR), vincristine (target: tubulin), and bortezomib (PSMB5 and PSMB1). While in the original dataset, samples were analyzed in 8 fractions, for obtaining deep datasets samples were fractionated into 16 (A549), 23 (MCF-7), or 24 (RKO), and the resulting fractions were analyzed using the Q Exactive HF mass spectrometer. The depths of the proteome profiles are 7398, 8735, and 8551 proteins, respectively, with no missing values in all three replicates. The comparison of number of proteins, number of peptides, average sequence coverage, and the number of missing values is given in Supplementary Fig. 11.

To showcase the applicability of the new deep datasets in target deconvolution, OPLS-DA models were built for the kinase inhibitor dasatinib vs. eight other treatments in each cell line (Fig. 6a). Comparison of the models revealed both similarities and cell-specific differences. As an example, the known target MAPK14 and the previously unknown target candidate PARG (Poly(ADP-ribose) glycohydrolase) were among the top target candidates in all three cell lines, while the other known kinase targets only appeared in certain cell lines (Fig. 6a). Dasatinib potently inhibits several tyrosine kinases, of which only 4 kinases including SRC, YES1, CSK, and LYN were among the top proteins in different cell lines. This is while the OPLS-DA model highlights other previously unknown kinases as potential targets. Interestingly, CYP1A1 which is involved in dasatinib metabolism[50], was the top up-regulated protein in MCF-7 cells (this protein was not quantified in A549 or RKO cells) (Fig. 6a). Similarly, the known targets for bortezomib dipeptidyl peptidase 2 (DPP7) and DPP3[51] were among the top proteins in MCF-7 cells (ranking 1 and 5 as down-regulated proteins) (Fig. 6b), but not in the other two cell lines. These datasets could provide a platform for merging user data.

**Merging deep datasets to obtain common drug targets and MOA.** In a previous study we have shown that a merged OPLS-DA model built for the proteomes of three cell lines perturbed by different drugs possesses an interesting property – it is rigid, meaning that such models created for a subset encompassing

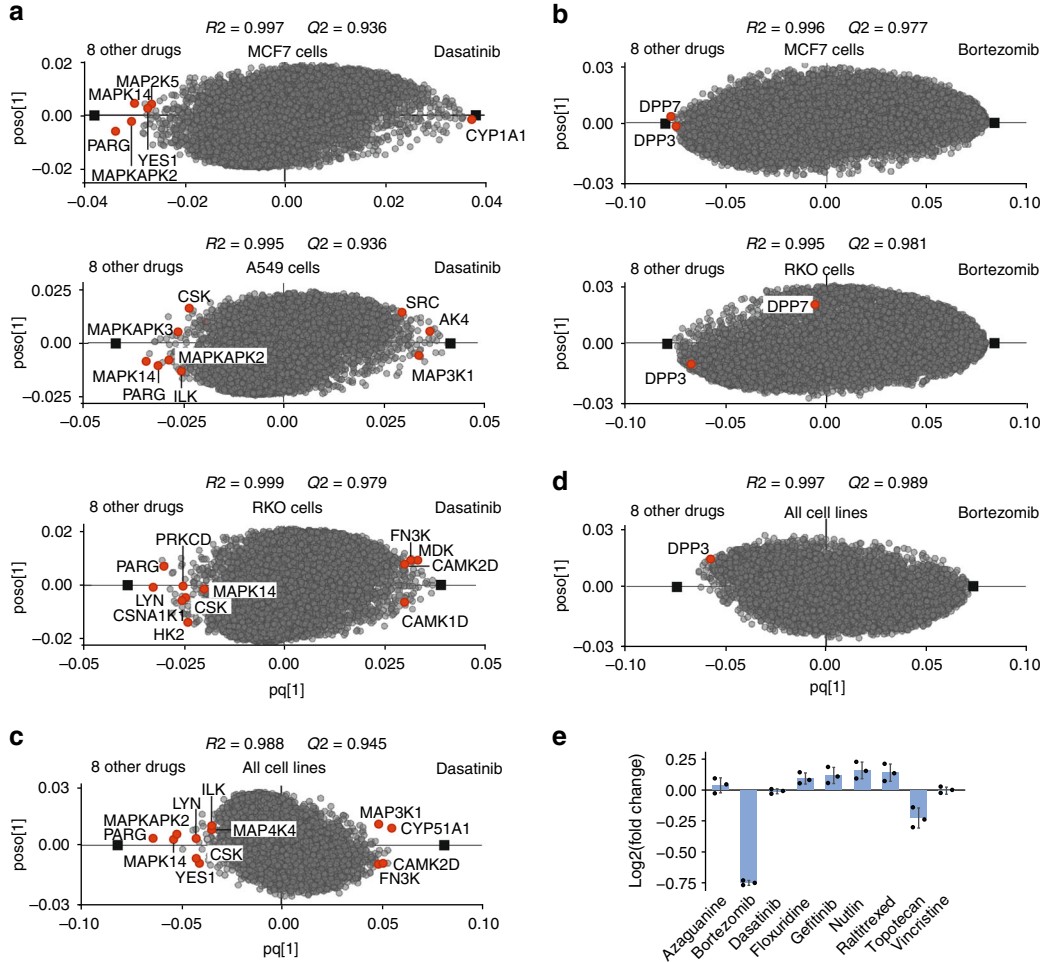

**Fig. 6 Deeper ProTargetMiner dataset with 8 contrasting compounds is successful in target/MOA deconvolution. a** OPLS-DA enabled deconvolution of multiple kinases as targets for dasatinib in three cell lines (drug targets shown in red circles). **b** deconvolution of DPP7 and DPP3 in MCF-7 and DPP3 in RKO cells. In A549 cells, these targets were not among the top 100 proteins. **c** identification of multiple targets for dasatinib in the merged dataset from three cell lines. **d** merging all cell lines shows that DPP3 is a common bortezomib target. Moreover, protein ubiquitination was the top GO term for 30 up-regulated proteins for bortezomib (11/30 proteins, $p < 2E-06$) in the merged dataset. **e** the specific down-regulation of DPP3 in response to bortezomib in MCF-7 cells. Data are represented as mean ± s.d. ($n = 3$ biologically independent experiments). Supplementary Data 4–7 were used for making panels **a**–**e**. Source data are provided as a Source Data file.

≥500 most abundant proteins are very similar to the models encompassing ≈100 most drug-specific proteins[52]. This is in a sharp contrast to single-cell models that do not possess such a property, and thus are loose. Model rigidity is a desired property allowing one to deduce a common drug target[53]. To obtain a rigid ProTargetMiner model, we performed OPLS-DA modeling for the merged three cell line dataset.

The combined deep-proteome dataset has a total depth of 11,293 proteins quantified with at least two peptides, of which 6496 proteins were common in all cell lines and without missing values (Supplementary Data 7). Compared to individual cell datasets, the merged model built for these proteins possesses higher specificity. For instance, for dasatinib, the known targets become more significant outliers (Fig. 6c).

On the other hand, some well-known targets faded in ranking, likely due to the cell-specific mechanistic differences. This phenomenon was noted for some kinases for dasatinib (Fig. 6c) and for DPP7 and DPP3 proteins for bortezomib (Fig. 6d). As an example, the expression of DPP3 in MCF-7 cells has been shown in Fig. 6e. Similarly, tubulins which had high rankings in MCF-7 and A549 cells in response to vincristine (3 tubulins were among the top 10 specifically down-regulated proteins in both cell lines),

were not among the top proteins in RKO cells. As a result, top proteins for vincristine in the merged dataset had only two tubulins. On the other hand, the top 30 specifically down-regulated proteins for vincristine in the merged dataset, mapped very well to rRNA processing ($p < 4E-5$) GO process and to the nucleolus ($p < 9E-6$) component, in agreement with the known vincristine effect on RNA synthesis[54]. Thus, the merged dataset presents two diverse MOAs of vincristine.

**Making an expandable public platform.** The ProTargetMiner datasets can be easily extended with new data on other compounds. In order to make the resource available to the community, a Shiny package was written in R, providing a user interface for data integration and PLS-DA modeling for either a selected cell line or all cell lines (Fig. 7a). In short, the user obtains the proteome signature of the desired compound at LC50 concentration (48 h treatment) in any, or all, of the above cell lines in form of the gene names and fold changes in preferably three replicates (only 6 proteomics analyses per cell line), and uploads this information (as an own dataset) through a user interface according to the step-by-step procedure given in Fig. 7b. A

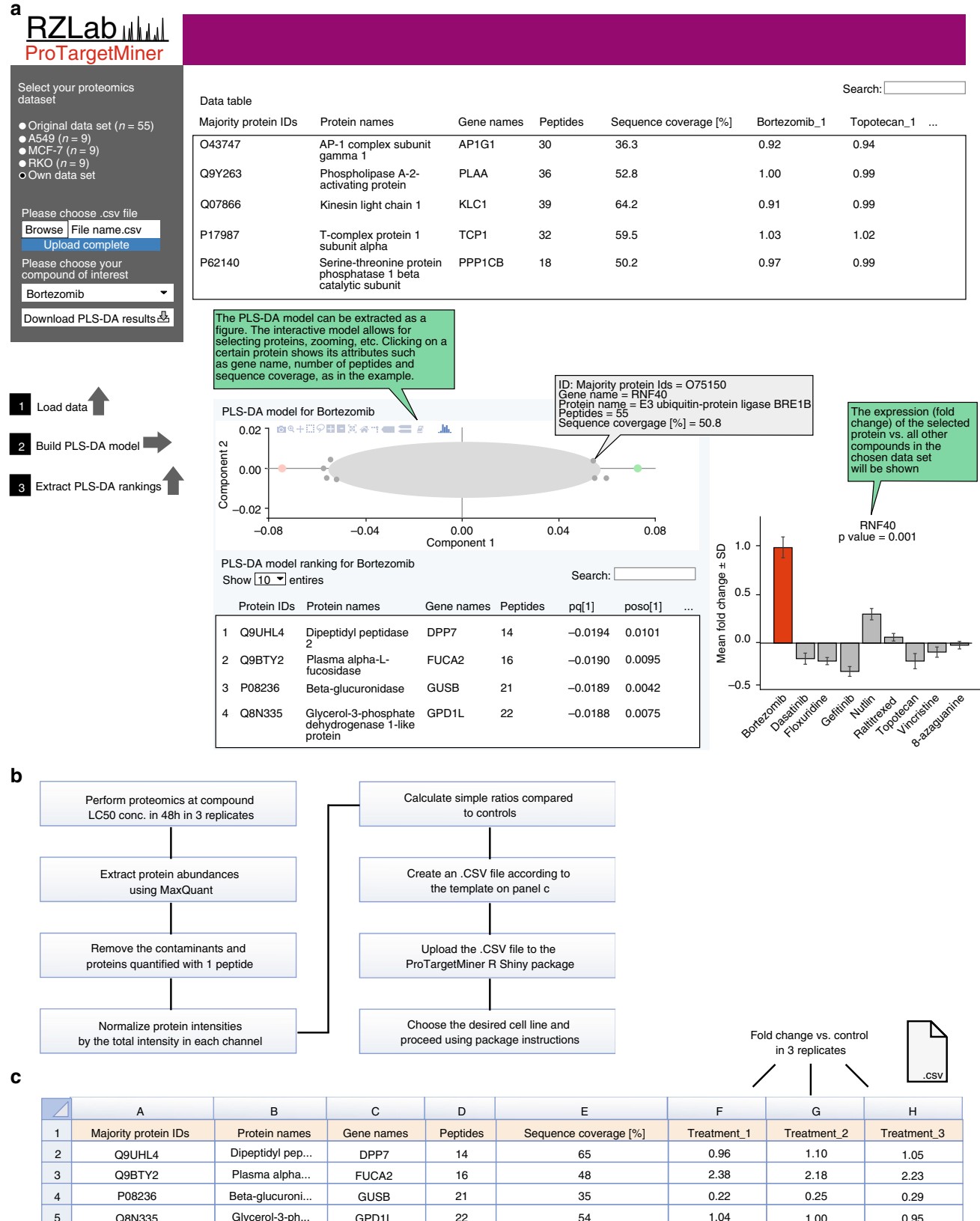

**Fig. 7 The ProTargetMiner R Shiny package for deconvolution of drug target and MOA. a** the ProTargetMiner R Shiny package interface. The input is the proteomics data for a desired compound and the output is the .CSV file containing the ranking of proteins for the compound against the desired panel in a PLS-DA model. Clicking on the interactive PLS-DA plot gives the attributes of the selected proteins, e.g. name, number of peptides, sequence coverage and significance compared to control, and will show the regulation of that protein in the compound panel. **b** step-by-step procedure of the use of ProTargetMiner R Shiny package. **c** the input .CSV template with the required columns.

template for the input.CSV file is shown in Fig. 7c. The package output will be the loading values of top specifically regulated proteins extracted from a PLS-DA model contrasting the given compound against the 9 diverse proteome signatures in that cell line (or all cell lines).

## Discussion

We generated an installment of a proteome signature library for anticancer molecules at LC50 concentrations, and proposed a modeling scheme implemented in an R Shiny package for deconvolution of drug targets, drug metabolizing enzymes, MOA, resistance factors and overall cellular effects for new compounds. We consider ProTargetMiner as a complement to preceding drug target deconvolution databases, such as connectivity maps[3,4,6]. The advantage of ProTargetMiner is that the response to anticancer molecules can be analyzed in detail on the protein level. Furthermore, the biological endpoint in this study is normalized, which makes the comparison of compound signatures more meaningful. The OPLS-DA modeling used in ProTargetMiner can be hypothetically applied to transcriptomics data or to non-anticancer treatments, as long as the biological endpoints for all used compounds are similar. We limited our discussion to the cases in which a single drug is contrasted against others, but the same approach can also be applied to characterize features shared among a selected class of compounds, or between any combinations of drugs. Such a methodology can also be applied to a panel of cell lines, e.g., inquiring which proteins specifically respond to a compound in a given cell line but not in other cell types.

Although the cytotoxicity of anticancer compounds was normalized in this study, it should still be noted that specific drugs might be imperfectly represented by proteomics profiles. For example, in the absence of expression of the primary targets in the cell system under study, off-target effects might dominate in the proteomic signature. Finding a single-cell system where all compounds can be tested in a specific way is very difficult if not impossible.

ProTargetMiner provided information on various aspects of different kinase inhibitors. One of the main challenges with kinase inhibitors is the high number of missing values in shotgun proteomics, especially when a large number of datasets are combined. Kinases are not among the most abundant cellular proteins. In the deep datasets we have tried to overcome this issue. While other types of enzymes can be hypothetically regulated by expression, the redundancy of function in most kinases results in insensitivity to differential regulation. It is yet to be seen what fraction of kinase inhibitor targets can be deconvoluted using ProTargetMiner.

The block-by-block and open source nature of this resource allow for expansion in various dimensions, e.g., by incorporating more perturbations, time points, and profiling more cell lines. Although expansion of the compound library seems desirable, one must consider that for a comprehensive database, enough perturbations must be done to saturate all possible cellular states. Ideally, highly specific inhibitors of every cellular protein are required. But given the astronomical number of potential perturbants, building a truly comprehensive library is a formidable project. While precision medicine targets specific cell types with defined mutations, building a comprehensive proteome response database for every such cell type seems impossible. Fortunately, as we have shown, ProTargetMiner approach can be easily customized and miniaturized. With top-of-the-line proteomics instruments reaching the depth of ≥10,000 proteins[55], a triplicate analysis of 9 perturbations requires less than a week of instrumental time. We hope that ProTargetMiner will be used by broad community of cancer researchers.

## Methods

**Compounds.** The library was cherry picked for cancer indication from a larger Selleckchem collection. The AXL inhibitor TP-0903 was from Selleckchem (Cat#S7846) and auranofin was from Sigma (Cat#A6733).

**Cell culture.** Human A549 cells (RRID:CVCL_0023; Cat#ATCC CCL-185; established from lung carcinomatous tissue from a 58-year-old Caucasian male), MCF-7 (RRID:CVCL_0031; Cat#ATCC HTB-22; established from breast adenocarcinoma from a 69-year-old Caucasian female), and RKO (RRID:CVCL_0504; Cat#ATCC CRL-2577; colon carcinoma cell line) (all obtained from ATCC, USA), were grown in DMEM medium (Fisher Scientific; Cat#11625200) supplemented with 10% FBS (Fisher Scientific; Cat#11560636), 2 mM L-glutamine (Fisher Scientific; Cat#BE17-605E) and 100 units per mL of penicillin/streptomycin (Thermo Fisher; Cat#15140122) and incubated at 37 °C in 5% CO₂. HFF-1 cells (RRID:CVCL_3285; Cat#ATCC SCRC-1041) were cultured under the exact same conditions in IMDM (Biowest). Cells were routinely tested for mycoplasma by MycoAlert Mycoplasma Detection Kit (#Cat: 11650261, Thermo Fisher Scientific). No authentication was performed, since passage number 2 cells were thawed and used from the ATCC source in all the experiments. In LC50 determination, cells were seeded at a density of 4000 per well in 96-well plates and after a day of growth, treated with the molecules for 48 h. Thereafter cell viability was measured using CellTiter-Blue® Cell Viability Assay (Promega; Cat#G8081) according to the manufacturer protocol.

**Proteomics.** For proteomics analysis, the cells were seeded at a density of 250,000 per well and allowed to grow for 24 h in biological triplicates. Next, cells were either treated with vehicle (DMSO) or compounds at LC50 concentrations. Each 10 experiments included one vehicle-treated control, 3 control drugs, and 6 library compounds. After treatment, cells were collected, washed twice with PBS (Fisher Scientific; Cat#11629980) and then lysed using 8 M urea (Sigma; Cat#U5378), 1% SDS, and 50 mM Tris at pH 8.5 with protease inhibitors (Sigma; Cat#05892791001). The cell lysates were subjected to 1 min sonication on ice using Branson probe sonicator and 3 s on/off pulses with a 30% amplitude. Protein concentration was then measured for each sample using a BCA Protein Assay Kit (Thermo; Cat#23227). 50 µg of each sample was reduced with DTT (final concentration 10 mM) (Sigma; Cat#D0632) for 1 h at room temperature. Afterwards, iodoacetamide (IAA) (Sigma; Cat#I6125) was added to a final concentration of 50 mM. The samples were incubated in room temperature for 1 h in the dark, with the reaction being stopped by addition of 10 mM DTT. After precipitation of proteins using methanol/chloroform, the semi-dry protein pellet was dissolved in 25 µL of 8 M urea in 20 mM EPPS (pH 8.5) (Sigma; Cat#E9502) and was then diluted with EPPS buffer to reduce urea concentration to 4 M. Lysyl endopeptidase (LysC) (Wako; Cat#125-05061) was added at a 1: 100 w/w ratio to protein and incubated at room temperature overnight. After diluting urea to 1 M, trypsin (Promega; Cat#V5111) was added at the ratio of 1: 100 w/w and the samples were incubated for 6 h at room temperature. Acetonitrile (Fisher Scientific; Cat#1079-9704) was added to a final concentration of 20% v/v.

TMT10 reagents (Thermo; Cat#90110) were added 4x by weight to each sample, followed by incubation for 2 h at room temperature. The reaction was quenched by addition of 0.5% hydroxylamine (Thermo Fisher; Cat#90115). Samples were combined, acidified by trifluoroacetic acid (TFA; Sigma; Cat#302031-M), cleaned using Sep-Pak (Waters; Cat#WAT054960) and dried using a DNA 120 SpeedVac™ concentrator (Thermo).

Samples were then resuspended in 0.1% TFA, and separated into 8 fractions using High pH Reversed-Phase Peptide Fractionation Kit (Thermo; Cat#84868). After resuspension in 0.1% FA (Fisher Scientific), each fraction was analyzed with a 90-min gradient in randomized order.

The deep proteomics samples (tags assigned in Supplementary Table 2) were prepared according to the above protocol until the multiplexing, cleaning, and drying steps, after which the samples were resuspended in 20 mM ammonium hydroxide and separated into 96 fractions on an XBridge BEH C18 2.1 × 150 mm column (Waters; Cat#186003023), using a Dionex Ultimate 3000 2DLC system (Thermo Scientific) over a 48 min gradient of 1–63%B (B = 20 mM ammonium hydroxide in acetonitrile) in three steps (1–23.5%B in 42 min, 23.5–54%B in 4 min and then 54–63%B in 2 min) at 200 µL min⁻¹ flow. Fractions were then concatenated into 16 samples in sequential order for A549 cells (e.g. 1, 17, 33, 49, 65, 81) and in 23 and 24 fractions for MCF-7 and RKO (e.g. 1, 25, 49, 73). After drying and resuspension in 0.1% formic acid (FA) (Fisher Scientific), each fraction was analyzed with a 90 min gradient (total method time = 110 min) in random order.

**LC-MS analysis.** Samples were loaded with buffer A (0.1% FA in water) onto a 50 cm EASY-Spray column (75 µm internal diameter, packed with PepMap C18, 2 µm beads, 100 Å pore size; Cat#ES803) connected to the EASY-nLC 1000 (Thermo; Cat#LC120) and eluted with a buffer B (98% ACN, 0.1% FA, 2% H₂O) gradient from 2 to 35% of at a flow rate of 250 nL min⁻¹. Mass spectra were acquired with an Orbitrap Q Exactive Plus mass spectrometer (Thermo; Cat# IQLAAEGAAPFALGMBDK) in the data-dependent mode with MS1 scan at 70,000 resolution, and MS2 at 35,000, in the m/z range from 375 to 1400. Peptide

fragmentation was performed via higher-energy collision dissociation (HCD) with energy set at 35 NCE.

For deep proteomics sets, samples were loaded with buffer A (0.1% FA in water) onto a 50 cm EASY-Spray column (75 μm internal diameter, packed with PepMap C18, 2 μm beads, 100 Å pore size) connected to a nanoflow Dionex UltiMate 3000 UPLC system (Thermo) and eluted in an increasing organic solvent gradient from 2 to 26% (B: 98% ACN, 0.1% FA, 2% $H_2O$) at a flow rate of 300 nL min$^{-1}$. Mass spectra were acquired with a Q Exactive HF mass spectrometer (Thermo; Cat#IQLAAEGAAPFALGMBFZ) in the data-dependent mode with MS1 scan at 120,000 resolution, and MS2 at 60,000 (@200 $m/z$), in the mass range from 350 to 1500 $m/z$. Peptide fragmentation was performed via higher-energy collision dissociation (HCD) with energy set at 33 NCE.

**Protein identification and quantification**. The raw data from LC-MS were analyzed by MaxQuant, version 1.5.6.5 (RRID:SCR_014485)[56]. The Andromeda engine[57] searched MS/MS data against Uniprot complete proteome database (human, version UP000005640_9606, 92957 entries). Cysteine carbamidomethylation was used as a fixed modification, while methionine oxidation and protein N-terminal acetylation were selected as a variable modification. Trypsin/P was selected as enzyme specificity. No more than two missed cleavages were allowed. A 1% false discovery rate was used as a filter at both protein and peptide levels. First search tolerance was 20 ppm (default) and main search tolerance was 4.5 ppm (default), and the minimum peptide length was 7 residues. Match between runs was activated with a match time window of 0.7 min and alignment time window of 20 min.

**Statistics**. After removing all the contaminants, only proteins with at least two peptides were included in the final dataset. Protein abundances were normalized by the total protein abundance in each sample in deep datasets. In the original dataset, protein intensities in every experiment (sample set) were normalized to ensure same median intensity across all channels in all replicates. Then for each protein log2-transformed fold-changes were calculated as a log2-ratio of the intensity to the mean of all control replicates. As the last step, log2-ratio were normalized across the whole dataset to ensure close to zero median log-fold change. The latter normalization does not affect the number of differentially regulated proteins; it accounts for experimental variations and is preferred for high-dimensional mass spectrometry data[58]. The data distribution before and after median normalization, and before and after log2 median fold change normalization are shown in Supplementary Fig. 12a–d, respectively, demonstrating the stabilization of the median fold change for replicates. To show that the same normalization would work for a subset of the dataset, we randomly took the data from experiment 4 and repeated the same procedure. A comparison of the data distribution in this subset from the original dataset and the individually normalized data is shown in Supplementary Fig. 13, demonstrating very similar results.

Data were processed by Excel, R, Python, and SIMCA (Version 15, UMetrics, Sweden; RRID:SCR_014688). All reported $p$ values are from two-sided Student's $t$-test (for cholesterol and AXL results).

**Network mapping**. GO pathway enrichment analysis for the top proteins derived from the OPLS-DA models was performed using the Gene Ontology enRIchment anaLysis and visuaLizAtion (GORILLA, http://cbl-gorilla.cs.technion.ac.il/) tool[59]. All the proteins quantified in each experiment were used as the background.

**Cholesterol quantitation**. Cellular total cholesterol was measured using cholesterol quantitation kit from Sigma (Cat#MAK043-1KT). In brief, 300k A549, RKO and HFF-1 cells were cultured in 6-well plates and treated with compounds for 20 h at 4 μM. Cholesterol was extracted by adding 200 μL of chloroform:isopropanol: IGEPAL CA-630 (7:11:0.1) and sonicated for 1 min on ice using Branson probe sonicator and 3 s on/off pulses with a 30% amplitude. The samples were centrifuged at 13,000 × $g$ for 10 min to remove the insoluble material. The organic phase was transferred to a new Eppendorf and dried. The lipids were then dissolved in assay buffer. The reaction mix consisting of assay buffer, cholesterol probe, enzyme mix, and cholesterol esterase was added to the samples in 96-well flat-bottom plates. The reactions were incubated for 60 min at 37 °C in the dark and the absorbance was measured at 570 nm.

**Reporting summary**. Further information on research design is available in the Nature Research Reporting Summary linked to this article.

## Data availability
The LC-MS/MS raw data files and extracted peptides and protein abundances are deposited in the jPOST repository of the ProteomeXchange Consortium[60] under the dataset identifier PXD009775 (original ProTargetMiner data), PXD009644 (deep proteomics set for A549 cells) and PXD013134 (deep proteomics set for MCF-7 and RKO cells) with no restrictions. The extracted protein abundances data and relevant outputs of data analysis are provided in Supplementary Data 1–7. The source data underlying Figs. 2b-i, 4b, g–j, and 6e, and Supplementary Figs. 1, 2a–h, 4b, 5, 6a, b, 9c, 10b, and 11a–h are provided as a Source Data file. All other data are available from the corresponding author on reasonable request.

## Code availability
The ProTargetMiner R Shiny package is available in GitHub (https://github.com/RZlab/ProTargetMiner) with no access restrictions. ProTargetMiner is also directly available on this domain: http://protargetminer.genexplain.com

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

## Acknowledgements

We would like to acknowledge Marie Ståhlberg and Carina Palmberg for their assistance in LC-MS/MS analyses. We are also grateful to Elias S.J. Arnér for providing TRi-1 and TRi-2 compounds, Stig Linder for b-AP15 and Kaori Sakurai for OSW-1 and tomatine. We would like to thank Alexander Kel for hosting ProTargetMiner on geneXplain website and Kamilya Altynbekova for its integration. This work was supported by grants from Cancerfonden (CAN 2014/381 and 2016/546). Bo Zhang would like to acknowledge Cancerfonden for financial support (CAN 2017/1081). Open access funding provided by Karolinska Institute.

## Author contributions

The concept, project organization, training, resources, and funding acquisition by R.A.Z.; Methodology by R.A.Z., A.A.S., and A.C.; Investigation by A.A.S., P.S., U.G.T., E.S., A.V., and M.G.; Data analysis by A.A.S., C.M.B., A.C., and B.Z.; Data curation by A.A.S.; Visualization by A.A.S., C.M.B., A.C., and B.Z.; R Shiny package by C.M.B.; Writing - original draft by A.A.S. and R.A.Z.; Writing - review & editing by R.A.Z. and A.A.S.

## Competing interests

B.Z. is currently an employee of AstraZeneca. Alexey Chernobrovkin is currently an employee of Pelago Bioscience AB. The other authors declare no potential conflicts of interest.
