## [Peer Review File · Nature Communications]

Reviewers' comments:

Reviewer #1 (Remarks to the Author):

In the manuscript 'ProTargetMiner: a proteome signature library of anticancer molecules for functional discovery' Zubarev and co-workers present an initial proteome signature library for anticancer drugs and provide an R-based package for data analysis that provides clues about potential drug targets, mechanism of action and cellular response to drug treatment in general. This tool in essence mimics on the proteome level mRNA expression based tools such as connectivity map. The authors provide proteome maps for 56 anti cancer drugs in A549 cells which is a valuable resource on its own right and suggest a framework of analysis methods and tools to infer from proteome signatures to potential targets and mechanism of action. This is an interesting resource for cancer drugs that complements existing target databases such as drug bank and that can alert to potential additional targets and target- or off-target related secondary effects on the proteome. Examples are currently limited to anticancer drugs but the approach might be applicable to other indications provided that cell perturbations lead to a common biological endpoint (such as cell death).

There are a couple of points that the authors might want to consider when revising their manuscript.

Points:

- The resource could be more impactful if the authors would implement this into an easily accessible data base system that integrates future data by their own lab or from other researchers following the procedures outlined in the manuscript. Implementation into existing repositories such as drug bank or proteomicsdb might be an alternative to generating their own system.
- The discussion is a very nice and balanced reflection of what their approach can deliver. This balance is not consistently achieved in the results section that sometimes misses on reporting the absence of expected findings. For example, sorafenib is first and foremost a kinase inhibitor but none of its known kinase targets is detected in the OPLS-DA plots, also dasatinib was primarily designed as an (BCR-)Abl inhibitor and has been shown to very potently inhibit a large number of tyrosine kinases of which hardly any seems to show up in their plots.
- The authors should state stronger that specific anticancer drugs may be only very imperfectly represented in proteomics profiles despite a certain level of cytotoxicity. In cases where the primary target is not expressed or the cell system is not dependent on a certain pathway proteomics signatures might rather be dominated by off target effects. For example the efficacy of the targeted covalent BTK inhibitor ibrutinib in certain form of leukaemia will not be reflected in A549 lung cancer cells despite the fact that the compound will be cytotoxic at high concentrations.

- There is hardly any novelty on inhibitors reported in the study. An exception might be the finding of dysregulation of cholesterol metabolism. Unfortunately, there is little follow-up that plausibly explains the mechanism underlying this dysregulation
- I have not found a detailed description of the OPLS-DA modeling presented in this paper, nor an explanation of the acronym for that matter. All in all the description of statistical procedures is rather light touch.
- The tools presented do not seem to give an estimate of significance of individual observations, nor an estimation of false discovery rate. The authors rather pick some proteins of interest at the extremes of the x-axis in their OPLS-DA plots but do not comment many other proteins with similar values. There seem to be no stringent thresholds or other heuristic criteria for picking the proteins of interest either.
- The section on the minimal size of the target miner system is a bit unclear. In figure 4 would it not be more useful to plot the median of all proteins with an indicator of spread rather than two randomly selected proteins that are not affected. Can the authors suggest an optimal initial set of compounds covering a wide variety of mechanisms to base the system on. There should be an influence of mechanistic diversity on the minimal size of the system.

minor

- In some cases the authors could have taken more care to provide the reader the information needed to understand their rationale. For example in the paragraph discussing size of the target miner vs. depth of the proteomics data set, the authors state that they acquired deeper proteomics data sets without detailing how much deeper, what was done differently. Judging from the details in materials and methods, they essentially doubled the number of fractions that were analysed with the same gradient. This information would be useful in results along with an average number of proteins or unique peptides per sample to better understand the difference to the above. Similarly, the authors could at least provide some information about the cholesterol measurements in the results section.
- After presenting the main features of their system, the authors elaborate on opportunities to base the target miner system on less data and on the influence of the level of proteome coverage. It would be useful if the authors could include a concluding sentence for each of these sections stating the main messages.

Reviewer #2 (Remarks to the Author):

The authors have generated a proteome signature library for anticancer molecules at LC50 concentrations. The main dataset is based on 287 A549 adenocarcinoma proteomes affected by 56

compounds. In addition, the authors provide an R Shiny package to perform OPLS-DA modelling for a selected compound using the provided data or data by the user. While the proposed tool in principle might be a valuable tool for many researchers, it has some drawbacks that need to be addressed. Another limitation of the manuscript is lack of some details, which make the manuscript difficult to follow. Finally, the approach as such is not novel. The idea of extending the concept of connectivity maps to proteomics has already been presented (PMID 29655704; 90 drugs in 6 cell lines) and the same data analysis methodology has been used by the authors in their previous publications (e.g. PMID 29572246).

Major comments:

The OPLS-DA method used by the authors as a key component in the analysis to interpret protein regulation and drug specificity comes with its caveats. It is known that OPLS-DA can easily produce statistically unreliable group separation and is even sometimes used as an alternative method if for example PCA fails to separate the groups (PMID 27547730). If OPLS-DA is the chosen modelling technique, the authors should thoroughly validate their findings using e.g. permutation testing and cross-validation. This is an important step regardless the evidence that the authors found from literature for their "counter-intuitive results" (row 266).

The authors are performing multiple normalization methods sequentially for their data (rows 536-542) and should provide a rationale for this. For instance, when the log₂ protein ratios between the sample and all control samples are scaled to have zero median fold changes, there is a great possibility that much of the produced signal is biased. This is especially true if the samples are behaving very differently, which is often the case when cancer cell lines are treated with intense compounds.

More details should be provided on various aspects of the methodologies as well as some of the results: 1) Please include the PCA analysis results described in rows 143-145. 2) Please clarify what exactly are the 287 signatures mentioned in the abstract (how many samples of each type). 3) Please provide details about the technical quality of the datasets, including reproducibility between the replicates and the role of missing values. 4) Please provide details on how you determined tomatine as an outlier and show its behavior. 5) Please provide details on how you defined significant regulation (row 173) and significant outlier (row 236), including statistical test applied and p-value. 6) Please specify which statistical test was used to determine Gene Ontology and pathway enrichment analysis and justify why "30 most specifically up- or down-regulated proteins were selected" (rows 254-255). 7) Please clarify which results you mean by "All reported p values are from two-sided Student's t-test." and justify the assumption about normally distributed data (rows 543-544). 8) Please include in the text a short description of the OPLS-DA modelling and its implementation.

Minor comments:

The idea of applying connectivity map for proteomics, while not completely novel, is nevertheless interesting. Is it possible to cross-confirm the analysis results using data from the previous proteomics and mRNA studies?

Overall, the text is somewhat hard to follow. It seems to be best suited for experts knowing exactly this narrow topic.

The references appear to be broken in the Materials and Methods section.

Fig. 5f is missing, corresponding to figure legend "f, the enrichment of "poly(A) RNA binding" (13 proteins, $p=7.47E-06$) and "ribonucleoprotein complex biogenesis" (6 proteins, $p=0.04$) as down-regulated proteins in the merged dataset for vincristine. Data are represented as $\text{mean}\pm\text{SD}$."

Reviewers' comments:

Reviewer #1 (Remarks to the Author):

In the manuscript 'ProTargetMiner: a proteome signature library of anticancer molecules for functional discovery' Zubarev and co-workers present an initial proteome signature library for anticancer drugs and provide an R-based package for data analysis that provides clues about potential drug targets, mechanism of action and cellular response to drug treatment in general. This tool in essence mimics on the proteome level mRNA expression based tools such as connectivity map. The authors provide proteome maps for 56 anti cancer drugs in A549 cells which is a valuable resource on its own right and suggest a framework of analysis methods and tools to infer from proteome signatures to potential targets and mechanism of action. This is an interesting resource for cancer drugs that complements existing target databases such as drug bank and that can alert to potential additional targets and target- or off-target related secondary effects on the proteome. Examples are currently limited to anticancer drugs but the approach might be applicable to other indications provided that cell perturbations lead to a common biological endpoint (such as cell death). There are a couple of points that the authors might want to consider when revising their manuscript.

Response: Thanks for the fine analysis and appraisal of our manuscript.

Comment 1. The resource could be more impactful if the authors would implement this into an easily accessible data base system that integrates future data by their own lab or from other researchers following the procedures outlined in the manuscript. Implementation into existing repositories such as drug bank or proteomicsdb might be an alternative to generating their own system.

Response: The provided R Shiny package embeds the data described in the manuscript and allows users to expand the database or upload own datasets. Our data can also be easily merged with user data using the provided accessions for all the proteins. Furthermore, to fulfill the reviewer comment, ProTargetMiner will now be directly available online through this web page: <http://protargetminer.genexplain.com/>

Comment 2. The discussion is a very nice and balanced reflection of what their approach can deliver. This balance is not consistently achieved in the results section that sometimes misses on reporting the absence of expected findings. For example, sorafenib is first and foremost a kinase inhibitor but none of its known kinase targets is detected in the OPLS-DA plots, also dasatinib was primarily designed as an (BCR-)Abl inhibitor and has been shown

to very potently inhibit a large number of tyrosine kinases of which hardly any seems to show up in their plots.

Response: We tried to provide a similar balance in the result section. Unfortunately, missing values are the inherent drawback of shotgun proteomics, resulting in false negatives (missing some of the targets). This issue is more prominent when data from multiple experiments are combined (Supplementary Figure 4 shows how merging the experiments comes at the expense of more missing values). Furthermore, as also now discussed in further detail in the paper, kinase targets are not the most suitable to ProTargetMiner analysis, because their inhibition often does not lead to significant protein abundance changes, likely due to the redundancy of kinase activity. This shortcoming is however ameliorated in the deep data sets, where we find multiple targets for dasatinib. For dasatinib, in the merged deep data set, we uncover 4 known dasatinib targets (among 10 that were available in the data set) out of 23 targets in drugbank. Furthermore, we identify novel target candidates that have not been reported before. In different OPLS-DA plots in different cell lines, overall four known tyrosine kinases including SRC, YES1, CSK and LYN were identified. BCR-ABL was not among the top proteins.

Comment 3. The authors should state stronger that specific anticancer drugs may be only very imperfectly represented in proteomics profiles despite a certain level of cytotoxicity. In cases where the primary target is not expressed or the cell system is not dependent on a certain pathway proteomics signatures might rather be dominated by off target effects. For example the efficacy of the targeted covalent BTK inhibitor ibrutinib in certain form of leukaemia will not be reflected in A549 lung cancer cells despite the fact that the compound will be cytotoxic at high concentrations.

Response: We expanded the discussion section with this consideration.

Comment 4. There is hardly any novelty on inhibitors reported in the study. An exception might be the finding of dysregulation of cholesterol metabolism. Unfortunately, there is little follow-up that plausibly explains the mechanism underlying this dysregulation.

Response: As a proof of principle study, the current work aimed mostly at “discovering” already known targets, which validated the approach, while discovering truly novel targets was a secondary objective due to the difficulty and expense associated with validation of such targets. Apart from cholesterol-related findings, we have also provided data supporting that AXL up-regulation can be a resistance factor against sorafenib and regorafenib toxicity. Furthermore, we provide new insight in the action of some pyrimidine analogues, which are shown for the first time to induce ribosomal stress.

Comment 5. I have not found a detailed description of the OPLS-DA modeling presented in

this paper, nor an explanation of the acronym for that matter. All in all the description of statistical procedures is rather light touch.

Response: OPLS-DA is now described in higher detail and clarity in the paper. A figure (number 3) was now added to elaborate on OPLS-DA. The statistical procedures are also explained better. VIP parameter is also discussed for prioritizing the targets.

Comment 6. The tools presented to not seem to give an estimate of significance of individual observations, nor an estimation of false discovery rate. The authors rather pick some proteins of interest at the extremes of the x-axis in their OPLS-DA plots but do not comment many other proteins with similar values. There seem to be no stringent thresholds or other heuristic criteria for picking the proteins of interest either.

Response: In OPLS-DA, the statistical variation between the replicates is automatically accounted for, with larger variations leading to a smaller x coordinate. Thus each protein on the extreme of x axis is of high potential interest. As we now explain, the statistical significance of each of these outliers can be estimated from the plots (using VIP values) an example of which is now shown in Figure 4. Furthermore, P-values against control for each data point is now provided in the R Shiny package. As for the false discovery rate, this isn't an easy issue because the terms "target" is only loosely defined, without any quantitative measure or threshold associated with it. The proteins of interest chosen to be shown are known for the selected drugs and are therefore, serving as proof-of-principle. It is practically impossible to validate or comment on all the outlying proteins.

Comment 7. The section on the minimal size of the target miner system is a bit unclear. In figure 4 would it not be more useful to plot the median of all proteins with an indicator of spread rather than two randomly selected proteins that are not affected. Can the authors suggest an optimal initial set of compounds covering a wide variety of mechanisms to base the system on. There should be an influence of mechanistic diversity on the minimal size of the system.

Response: The median of all proteins would be a horizontal line running at $N/2$, where N is the number of proteins. Regarding the optimal initial set of compounds covering a wide variety of mechanisms, the 9 drugs used for deep proteome analysis in three other cell lines serve as such a set, as these compounds were found to represent most orthogonal proteome responses (most diverse mechanisms).

Minor comments

Comment 8. In some cases the authors could have taken more care to provide the reader the information needed to understand their rationale. For example in the paragraph discussing size or the target miner vs. depth of the proteomics data set, the authors state that they acquired deeper proteomics data sets without detailing how much deeper, what

was done differently. Judging from the details in materials and methods, they essentially doubled the number of fractions that were analysed with the same gradient. This information would be useful in results along with an average number of proteins or unique peptides per sample to better understand the difference to the above. Similarly, the authors could at least provide some information about the cholesterol measurements in the results section.

Response: The required information was added regarding the deep proteome experiments in Supplementary Figures 4 and 9. The cholesterol results were further explained in the results section.

Comment 9. After presenting the main features of their system, the authors elaborate on opportunities to base the target miner system on less data and on the influence of the level of proteome coverage. It would be useful if the authors could include a concluding sentence for each of these sections stating the main messages.

Response: We thank the reviewer for this comment and added concluding sentences for most of these sections.

Reviewer #2 (Remarks to the Author):

Comment 1. The authors have generated a proteome signature library for anticancer molecules at LC50 concentrations. The main dataset is based on 287 A549 adenocarcinoma proteomes affected by 56 compounds. In addition, the authors provide an R Shiny package to perform OPLS-DA modelling for a selected compound using the provided data or data by the user. While the proposed tool in principle might be a valuable tool for many researchers, it has some drawbacks that need to be addressed. Another limitation of the manuscript is lack of some details, which make the manuscript difficult to follow. Finally, the approach as such is not novel. The idea of extending the concept of connectivity maps to proteomics has already been presented (PMID 29655704; 90 drugs in 6 cell lines) and the same data analysis methodology has been used by the authors in their previous publications (e.g. PMID 29572246).

Response: We thank the reviewer for the meticulous analysis of our manuscript. We are aware of the proteomics connectivity map paper (PMID 29655704) and have duly cited and discussed this paper. ProTargetMiner however, does not pursue a connectivity map approach. We believe that the novelty of ProTargetMiner is in using for every drug the concentration that causes an equivalent biological effect (LC50 at 48 h), which allows for more adequate mapping of the cell state, and more accurate determination of the targets and action mechanism.

Furthermore, unlike connectivity maps, ProTargetMiner provides specifically regulated proteins. Indeed, OPLS-DA modeling has been used in our previous publication (PMID 29572246), but here we have greatly increased the specificity by including many more

compounds. Besides, unlike our previous approach, ProTargetMiner database is easily expandable. For including a new compound, one needs to analyze only 6 proteomes from a single cell line.

Major comments:

Comment 2. The OPLS-DA method used by the authors as a key component in the analysis to interpret protein regulation and drug specificity comes with its caveats. It is known that OPLS-DA can easily produce statistically unreliable group separation and is even sometimes used as an alternative method if for example PCA fails to separate the groups (PMID 27547730). If OPLS-DA is the chosen modelling technique, the authors should thoroughly validate their findings using e.g. permutation testing and cross-validation. This is an important step regardless the evidence that the authors found from literature for their "counter-intuitive results" (row 266).

Response: We agree that statistical reliability of the OPLS-DA models is a very important issue, and have now provided detailed explanation of the evaluation of statistical uncertainty in protein OPLS-DA coordinates (see our reply to Comment 6 of Reviewer 1). This uncertainty derives from the variability between the three replicate analyses, and it is more reliable than permutation. The latter ignores such important parameter as the number of unique peptides used to quantify a given protein, while we used this parameter for picking up the most reliable proteins among those with similar x-coordinates. Cross-validation is not applicable here, as it would require orders of magnitude more replicates than three, which would be impossible to obtain. It should also be noted that here, OPLS-DA has been used to separate only two groups and the *OPLS-DA loadings* are shown and used for target deconvolution. Forced and unreliable separation with OPLS-DA usually refers to studies where multiple classes are separated. PCA would not be able to strictly separate two classes. Furthermore, now we provide p values against controls for every proteins on the plot in the R Shiny package. The validity of the findings can also be verified by checking the expression of the respective proteins in the data set.

Comment 3. The authors are performing multiple normalization methods sequentially for their data (rows 536-542) and should provide a rationale for this. For instance, when the log₂ protein ratios between the sample and all control samples are scaled to have zero median fold changes, there is a great possibility that much of the produced signal is biased. This is especially true if the samples are behaving very differently, which is often the case when cancer cell lines are treated with intense compounds.

Response: The biological effect of all compounds in our study was equivalent (50% viability after 48 h), thus in that respect all compounds were equally intense. It was therefore logical to assume that the total protein concentration per living cell was similar for all compounds, although some variations could of course occur. We did not explain this rational because of

the space limitations, but will gladly do that now with Editor's permission. Also, the users will have access to the data, and they will be able to change the normalization as they wish.

Comment 4. More details should be provided on various aspects of the methodologies as well as some of the results: 1) Please include the PCA analysis results described in rows 143-145.

Response: PCA results as well as the contribution of each component are now provided as Supplementary Figures 1 and 2.

Comment 5. 2) Please clarify what exactly are the 287 signatures mentioned in the abstract (how many samples of each type).

Response: This number is now clearly described.

Comment 6. 3) Please provide details about the technical quality of the datasets, including reproducibility between the replicates and the role of missing values.

Response: Relevant analyses and figures were provided in the paper (Supplementary Figure 3).

Comment 7. 4) Please provide details on how you determined tomatine as an outlier and show its behavior.

Response: This information was also added to the paper (please refer to Supplementary Figure 6).

Comment 8. 5) Please provide details on how you defined significant regulation (row 173) and significant outlier (row 236), including statistical test applied and p-value.

Response: This was further clarified in the paper. Extremely ranked proteins can be considered as outliers, as the specificity values change in each model. Furthermore, now we discuss VIP, which is an OPLS-DA parameter that can be used for ranking of target candidates. With our experience, up to 50 proteins from each extremity can be considered. We removed the word significantly, as only regulation was used as a measure to calculate the numbers in row 173. For the row 236, we have now included the p value and added details.

Comment 9. 6) Please specify which statistical test was used to determine Gene Ontology and pathway enrichment analysis and justify why "30 most specifically up- or down-regulated proteins were selected" (rows 254-255).

Response: The choice of 30 top proteins is empirical, as this number of proteins usually provides specific pathways. We have noted that including a higher number of proteins leads to redundancy of pathways and hampers deriving biological inferences. The p values are

provided by the String tool against the whole genome as the background, which is described in materials and methods.

Comment 10. 7) Please clarify which results you mean by “All reported p values are from two-sided Student’s t-test.” and justify the assumption about normally distributed data (rows 543-544).

Response: The t-tests were only performed for the cholesterol analysis and AXL as a resistance factor. This is now clarified in materials and methods.

Comment 11. 8) Please include in the text a short description of the OPLS-DA modelling and its implementation.

Response: As also requested by the other reviewer, the OPLS-DA modeling is now described in full detail in the newly added Figure 3.

Minor comments:

Comment 12. The idea of applying connectivity map for proteomics, while not completely novel, is nevertheless interesting. Is it possible to cross-confirm the analysis results using data from the previous proteomics and mRNA studies?

Response: We have investigated such a possibility and found that a detailed comparison would justify a separate study (with own data). A major problem is finding data where the compound LC50s, duration of treatment (48h) and cell lines are similar.

Comment 13. Overall, the text is somewhat hard to follow. It seems to be best suited for experts knowing exactly this narrow topic.

Response: We acknowledge that and have now tried to add more clarity to the text.

Comment 14. The references appear to be broken in the Materials and Methods section.

Response: We have double checked and corrected all the references.

Comment 15. Fig. 5f is missing, corresponding to figure legend “f, the enrichment of “poly(A) RNA binding” (13 proteins, $p=7.47E-06$) and “ribonucleoprotein complex biogenesis” (6 proteins, $p=0.04$) as down-regulated proteins in the merged dataset for vincristine. Data are represented as $\text{mean} \pm \text{SD}$.”

Response: Indeed; thanks for noting this issue. This caption was related to Figure 5e and the caption has been duly corrected.

Reviewers' comments:

Reviewer #1 (Remarks to the Author):

In the amended manuscript the authors have adequately addressed my initial criticism.

I have only two minor comments:

in lines 153-156 there seems to be a slight duplication, the authors will surely find a more concise way for expressing this point.

In lines 448-453 the authors have word-by-word repeated a comment I made in my initial referee report. Unless the authors want to add a reference to my initial report, I suggest the authors find their own words.

Reviewer #2 (Remarks to the Author):

The authors have addressed many of my comments. However, some of them would still need clarification. Overall, I find the extendable database itself a valuable resource, while the analysis is lacking in some aspects. Moreover, as stated before, the overall idea is not completely novel and the same methodology has been used by the authors before. The main points that require further clarification are listed below.

1. OPLS-DA validation: The authors still ignore the possibility of drawing false conclusions from their modelling approach and choose not to perform any kind of cross validation or permutation analysis. While arguably cross validation can be problematic using just three replicates, proper assessment of the reliability and significance of the results should be provided together with a detailed description of the approach. The least the authors should do is to show the expression levels of the selected individual proteins across all the replicates and describe in detail how the "significance compared to control" was calculated. It would also be important to analyse what happens if the same methodology is applied to data that does not contain any true signal.

2. Normalisation: While the authors argue that the compounds were equally intense (LC50), it would be surprising if their effects to the proteome would be equal. This raises a concern with the selected unconventional normalisation approach used in the analysis. The authors should clearly show how the data behaves both before and after normalisation, showing e.g. correlations and distributions. For example, what would happen if one of the compounds induces considerable upregulation in the proteome compared to the other compounds? Similarly, would the conclusions hold if only a subset of the compounds would be normalised and analysed at a time?

3. Enrichment analysis: The authors mention that the choice of a fixed number of “top 30 proteins in either side of x axis is empirical and leads to less redundancy in pathways”. However, no data was provided to support that. Also, if the whole genome was used as a background, it is likely that with the limited measurement capabilities of proteomics, many of the findings are just related to the tissue in question.

4. Tomatine: I still have difficulties understanding the behaviour of tomatine. The authors state that “We found tomatine to be a gross outlier in t-SNE as well as in PCA”. However, if one looks at the t-SNE plot (Supplementary Fig. 5), tomatine does not seem to be an outlier. In the PCA plot (Supplementary Fig. 2), I was not able to find tomatine at all. As described by the authors, tomatine seems to have a larger number of differentially expressed proteins than the other compounds. This leads to a similar question as with the normalisation: should one always expect approximately the same number of proteins to be affected? This needs clarification. Also, details should be provided on how the differential expression was defined.

Response to reviewer comments

Reviewer #1 (Remarks to the Author):

In the amended manuscript the authors have adequately addressed my initial criticism.

I have only two minor comments:

Comment 1: in lines 153-156 there seems to be a slight duplication, the authors will surely find a more concise way for expressing this point.

Response: The duplication was erased.

Comment 2: In lines 448-453 the authors have word-by-word repeated a comment I made in my initial referee report. Unless the authors want to add a reference to my initial report, I suggest the authors find their own words.

Response: We thank the esteemed reviewer for this comment. This paragraph was rewritten.

Reviewer #2 (Remarks to the Author):

The authors have addressed many of my comments. However, some of them would still need clarification. Overall, I find the extendable database itself a valuable resource, while the analysis is lacking in some aspects. Moreover, as stated before, the overall idea is not completely novel and the same methodology has been used by the authors before. The main points that require further clarification are listed below.

Comment 1. OPLS-DA validation: The authors still ignore the possibility of drawing false conclusions from their modelling approach and choose not to perform any kind of cross validation or permutation analysis. While arguably cross validation can be problematic using just three replicates, proper assessment of the reliability and significance of the results should be provided together with a detailed description of the approach. The least the authors should do is to show the expression levels of the selected individual proteins across all the replicates and describe in detail how the “significance compared to control” was calculated. It would also be important to analyse what happens if the same methodology is applied to data that does not contain any true signal.

Response:

We thank the esteemed reviewer for this valuable comment. OPLS-DA has an internal cross-validation process and generates a Q2 value. So while the R2 value represents model fit, Q2 is a measure of model predictive power, and it is calculated by cross-validation. Briefly, the data-set is divided into seven groups and subsequently each 1/7th is removed in turn. A model is built on the remaining 6/7th of the data and the left out data are predicted from the new model. This is repeated with each 1/7th until all the data have been predicted. The predicted data are then compared with the original data and the sum of squared errors is calculated for the whole dataset. This number is then converted into Q2 by dividing it by the initial sum of squares and subtracting from 1. Good predictions will have high Q2; a perfect prediction has Q2=1.

The R2 and Q2 values for all the presented models are now included in figures and the procedure is now fully described in the paper.

Moreover, following the reviewer comment, now we provide the expression levels of top proteins from the models in Figure 4. We had also included the expression of DPP3 for bortezomib in Figure 6, and the reviewer is correct that more examples must have been included to validate and reinforce the findings. The significance was calculated by two-sided Student t-test of the expression level in the treatment of interest to that of control. To show what happens when there is no signal, we removed

the compounds in Figure 4 from the data set (3 cases) and built OPLS-DA models with 3 randomly chosen columns instead (Supplementary Figure 8). The protein targets highlighted in Figure 4 disappeared from the top ranking list, indicating that random selection of columns does not support meaningful findings.

Comment 2. Normalisation: While the authors argue that the compounds were equally intense (LC50), it would be surprising if their effects to the proteome would be equal. This raises a concern with the selected unconventional normalisation approach used in the analysis. The authors should clearly show how the data behaves both before and after normalisation, showing e.g. correlations and distributions. For example, what would happen if one of the compounds induces considerable upregulation in the proteome compared to the other compounds? Similarly, would the conclusions hold if only a subset of the compounds would be normalised and analysed at a time?

Response: The reviewer is correct, the effect of compounds on the proteome is not equal, which we now also discuss in detail. In any meaningful normalization, the majority of proteins are unchanged, with the median expression level centering on zero on the log scale. We make sure that this is the case and provide Supplementary Figure 7 showing the extent of proteome disturbance for each compound (the main fully normalized dataset was used for generation of this figure). The data behavior before and after normalization was quite similar, as now shown in Supplementary Figures 12 and 13. The correlation of data before and after normalization for each replicate is 1. The normalization used in this paper is routine for big proteomics data and is only used for median normalization, and not for adjusting the number of differentially regulated proteins.

Comment 3. Enrichment analysis: The authors mention that the choice of a fixed number of “top 30 proteins in either side of x axis is empirical and leads to less redundancy in pathways”. However, no data was provided to support that. Also, if the whole genome was used as a background, it is likely that with the limited measurement capabilities of proteomics, many of the findings are just related to the tissue in question.

Response: The reviewer is correct about the complications in using whole genome as background. To address this comment, now we submitted the proteins to GORILLA with quantified proteins in each dataset as the background. The old pathway analysis results were replaced with new ones throughout the paper. The results of pathway analysis were highly similar. We used the 30 top hits for each drug for pathway analysis and GO term analysis as a quick look at the perturbed pathways and for hypothesis generation. Proper characterization of each individual compound is out of the scope of present manuscript. Furthermore, the ranking of all proteins for each compound are listed in Supplementary Data for everyone.

Comment 4. Tomatine: I still have difficulties understanding the behaviour of tomatine. The authors state that “We found tomatine to be a gross outlier in t-SNE as well as in PCA”. However, if one looks at the t-SNE plot (Supplementary Fig. 5), tomatine does not seem to be an outlier. In the PCA plot (Supplementary Fig. 2), I was not able to find tomatine at all. As described by the authors, tomatine seems to have a larger number of differentially expressed proteins than the other compounds. This leads to a similar question as with the normalisation: should one always expect approximately the same number of proteins to be affected? This needs clarification. Also, details should be provided on how the differential expression was defined.

Response: The reviewer is correct, and tomatine is only seen as an outlier in PCA. t-SNE will not show the outliers, as it is a dimension reduction technique for better visualization, which tries to keep the groups close together. This was an unintentional error from authors and this sentence was duly corrected. Furthermore, we have now provided the PCA plot of the whole data set in Supplementary Figure 4. We did not include this panel in the previous submission, because tomatine was squeezing all the other compounds in the PCA plot and thus the separation of other compounds could not be seen on the plot. Normalizing the data to a median of zero does not affect the number of differentially regulated proteins, as explained above. The number of differentially regulated proteins for different compounds is different, as shown in Supplementary Figure 7 and indirectly in Supplementary Figure

4b. The differential regulation is now defined in the text and also in the Supplementary Figure 4 caption as “fold change vs. control >2 and < 0.5 ”.

REVIEWERS' COMMENTS:

Reviewer #2 (Remarks to the Author):

The authors have adequately addressed my comments.